# Pathogen-derived extracellular vesicles mediate virulence in the fatal human pathogen *Cryptococcus gattii*

Ewa Bielska [1], Marta Arch Sisquella[2], Maha Aldeieg[3], Charlotte Birch[1], Eloise J. O'Donoghue[1] & Robin C. May [1]

The Pacific Northwest outbreak of cryptococcosis, caused by a near-clonal lineage of the fungal pathogen *Cryptococcus gattii*, represents the most significant cluster of life-threatening fungal infections in otherwise healthy human hosts currently known. The outbreak lineage has a remarkable ability to grow rapidly within human white blood cells, using a unique 'division of labour' mechanism within the pathogen population, where some cells adopt a dormant behaviour to support the growth of neighbouring cells. Here we demonstrate that pathogenic 'division of labour' can be triggered over large cellular distances and is mediated through the release of extracellular vesicles by the fungus. Isolated vesicles released by virulent strains are taken up by infected host macrophages and trafficked to the phagosome, where they trigger the rapid intracellular growth of non-outbreak fungal cells that would otherwise be eliminated by the host. Thus, long distance pathogen-to-pathogen communication via extracellular vesicles represents a novel mechanism to control complex virulence phenotypes in *Cryptococcus gattii* and, potentially, other infectious species.

[1] Institute of Microbiology and Infection, School of Biosciences, University of Birmingham, Edgbaston, Birmingham B15 2TT, UK. [2] Institut d' Investigació en Ciències de la Salut Germans Trias i Pujol (IGTP), Crta de Can Ruti s/n, Badalona 08916 Catalonia, Spain. [3] School of Biological Sciences, University of Reading, Knight Building, Whiteknights Campus, Reading RG6 6AJ, UK. Correspondence and requests for materials should be addressed to E.B. (email: ewabielska@mail.com) or to R.C.M. (email: r.c.may@bham.ac.uk)

Cryptococcosis is a major human and animal life-threatening fungal disease[1–3]. Globally, most human infections are caused by *Cryptococcus neoformans*, with the related species *Cryptococcus gattii* causing less than 1% of all human cryptococcal disease. However, in the late 1990s a near-clonal lineage of *C. gattii* became established in British Columbia, Canada, and subsequently caused a major cluster of human and animal disease that has come to be known as the Pacific Northwest Outbreak[4]. A defining feature of cells within the outbreak lineage is their ability to proliferate very rapidly within host phagocytes[5]. We previously demonstrated that this rapid proliferation is driven by a 'division of labour' mechanism, in which individual fungal cells coordinate their behaviour to maximise proliferation of the population as a whole[6]. However, the mechanism by which this coordination occurs at a cellular level has remained enigmatic.

Here we demonstrate that the key regulator of this 'division of labour' process is the release and exchange of extracellular vesicles (EVs) by outbreak strains of *C. gattii*. These EVs are efficiently taken up by infected host macrophages and trafficked to the fungal phagosome, where they induce rapid proliferation of the recipient pathogen cell and thereby drive pathogenesis in this highly virulent lineage.

## Results

### 'Division of labour' can be triggered over large distances.

We previously showed that outbreak strains of *C. gattii* induce the rapid intracellular proliferation of otherwise non-virulent strains during co-infection[6]. To test whether this effect required the fungal cells to be present within the same host cell, we generated fluorescently tagged versions of an outbreak (R265[7]) and non-outbreak (ICB180) strain of *C. gattii* and confirmed that these strains were unaltered from their parental strains in morphology, growth or stress tolerance (Supplementary Fig. 1). ICB180-mCherry and R265-GFP were used to infect J774 macrophages either alone or together. Microscopy observations of infected macrophages at 2 hours post infection (h.p.i.; Fig. 1a) and 24 h.p.i revealed that dually infected host cells were exceptionally rare (2/5479 infected host cells; Fig. 1a), suggesting that isolates do not need to be in the same phagocyte to trigger 'division of labour'. To test this in more detail we infected macrophages with the non-outbreak strain ICB180 and then used a transwell system (ThinCert™) to physically separate *C. gattii* R265 (an outbreak strain) from contact with the macrophages while continuing to allow them to freely exchange particles below the 400 nm transwell cut off (Fig. 1b). Under these conditions, the presence of an outbreak strain, but not a non-outbreak strain, in the upper

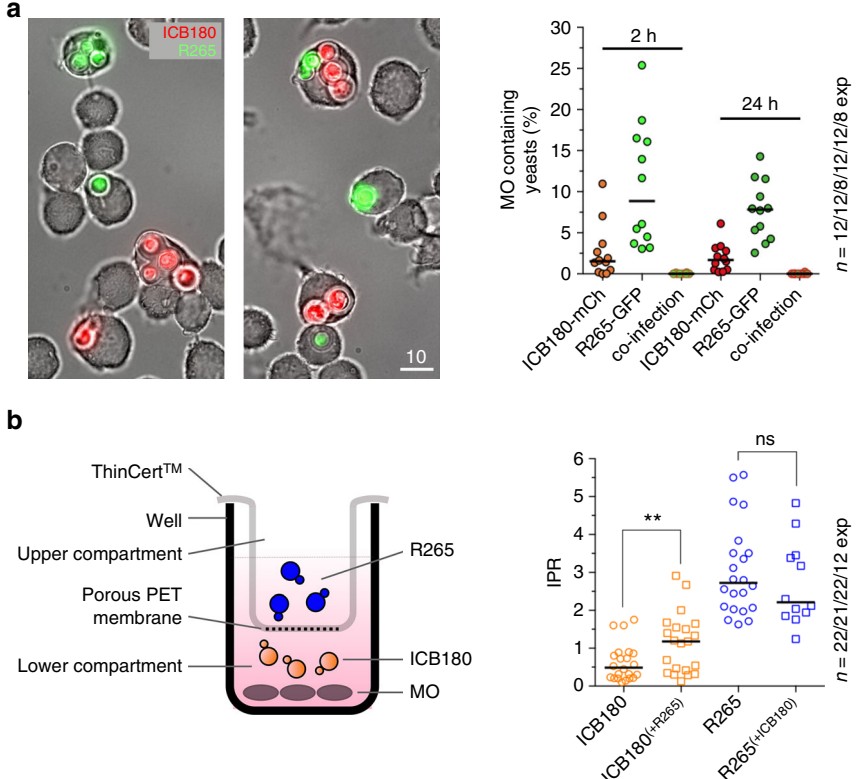

**Fig. 1** Long-distance communication can drive rapid intracellular proliferation in *C. gattii*. **a** During co-infection, different strains of *C. gattii* (R265-GFP shown in green; ICB180-mCherry shown in red) are rarely phagocytosed by the same macrophage. Bar: 10 μm. The number of infected macrophages containing both isolates of yeast at the same time is very low at 2 h.p.i. (2 in total 5479 tested macrophages) and at 24 h.p.i (1 in total 3235 tested macrophages). Data are presented as scattered dot plots with lines representing their medians. Data are representative of results from 8–12 independent experiments with a minimum of 150 macrophages analysed per sample per experiment. **b** A schematic representation of the experiment using transwell system ThinCert™ with 400 nm porous membrane that separates lower from upper compartments thereby allowing splitting of growth of two different *C. gattii* strains R265 (pathogenic) and ICB180 (non-pathogenic). After two initial hours of the infection the transwell system was removed and intracellular proliferation rate (IPR) of ICB180 was measured (as $T_0$) and after following 24 h (as $T_{24}$). The 2-h presence of R265 (outbreak) cryptococci in the transwell system (ICB180$^{(+R265)}$) induces significantly higher intracellular proliferation of ICB180 (non-outbreak strain) within macrophages ($P = 0.0038$, Wilcoxon matched-pairs signed rank test), an effect that is not seen when R265 is replaced for ICB180 (R265$^{(+ICB180)}$; $P = 0.9263$, Wilcoxon matched-pairs signed rank test). Data are presented as scattered dot plots with lines representing their medians. Data are representative of results from 12–22 independent experiments with 879–5238 total yeasts counted for each sample. Wilcoxon matched-pairs signed rank test where ** ($P \le 0.01$), significant difference; ns ($P > 0.05$), not different

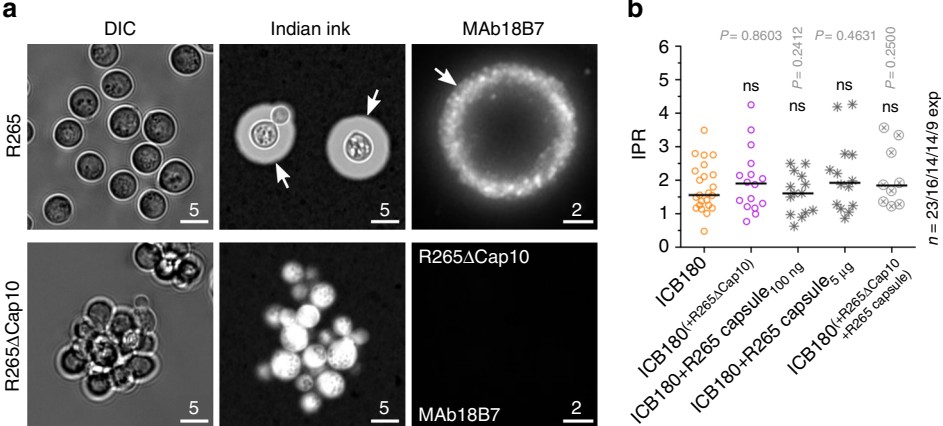

**Fig. 2** Capsular material is necessary but not sufficient to increase the IPR of non-outbreak cryptococci. **a** Differential interference contrast (DIC; left), India Ink (middle) and immunostained (MAb 18B7; right) images of wild-type R265 (top) and acapsular R265ΔCap10 (bottom), indicating the presence of the characteristic thick polysaccharide capsule in the wild-type (arrows). Note the lack of any visible capsule in the acapsular mutant. Bars: 5 and 2 μm. **b** IPR of a non-virulent ICB180 strain is not altered by the presence of acapsular R265ΔCap10 in the transwell assay (ICB180$^{(+R265\Delta Cap10)}$), nor by the addition of capsular material isolated from R265 (ICB180+R265 capsule$_{100\text{ ng}}$; ICB180+R265 capsule$_{5\text{ μg}}$). An addition into the transwell of acapsular R265ΔCap10 strain mixed with the capsular material isolated from R265 also did not change the IPR of ICB180 (ICB180$^{(+R265\Delta Cap10+R265\text{ capsule})}$). Data are presented as scattered dot plots with lines representing their medians. Individual Wilcoxon matched-pairs signed rank test presented as *P* values above each dot plot, where ns (*P* > 0.05), not significantly different. Data are representative of results from at least nine independent experiments with 808–1913 total number of yeasts counted for each sample

compartment of the transwell was sufficient to raise the intracellular proliferation rate (IPR) of ICB180 within macrophages by 2.5 fold (Fig. 1b). Thus released molecules from outbreak *C. gattii* are sufficient to recapitulate the 'division of labour' phenotype over large distances.

**Capsular material is not sufficient to induce higher IPR.** It is well documented that cryptococcal capsular material, which is shed during infection, is essential for replication inside macrophages[8] and can induce potent immunomodulatory effects on a range of leucocytes[9–11]. To test whether capsular material is responsible for triggering the rapid intracellular replication we observe in outbreak cryptococci we used a non-capsular strain, R265ΔCap10[12] (Fig. 2a), instead of wild-type R265 in the transwell system experiment. Unlike wild-type R265, acapsular R265 was unable to induce increased proliferation of intracellular ICB180 in this system (Fig. 2b). However, adding purified capsular material derived from wild-type R265 was also unable to recapitulate the raised IPR of ICB180 (Fig. 2b and Supplementary Fig. 2b). Addition of capsular material to the acapsular strain R265ΔCap10 in vitro (Fig. 2b and Supplementary Fig. 2a) further suggested that the presence of an intact capsule secretion pathway is necessary to trigger raised IPRs in non-outbreak cryptococci.

***C. gattii* releases extracellular vesicles.** Recent findings from a diverse range of organisms have highlighted the importance of EVs in long-range cell-to-cell signalling and we therefore considered whether such vesicles may be produced by *C. gattii* and contribute towards 'division of labour'. EVs are known to be produced by the closely related species *C. neoformans*[13] but have not been previously identified in *C. gattii*. Using an ultracentrifugation method[13] we were able to purify EVs from the Pacific Northwest outbreak strain R265, the acapsular strain R265ΔCap10 and the non-outbreak strain ICB180. Electron microscopy and nanoparticle tracking analysis (NTA)-based measurement revealed EVs to have a typical spherical shape with a diameter of 100 nm or less (Supplementary Fig. 3a). Detailed NTA analysis (Supplementary Fig. 3b−d) demonstrated that EVs

from R265 were significantly larger (26−397 nm, median size of 108 nm) than those from R265ΔCap10 (median 72.8 nm) or ICB180 (median 86.3 nm), in agreement with data from *C. neoformans* showing reduced EV size in acapsular strains[14].

**Outbreak-derived EVs enhance the IPR of non-outbreak strains.** To test whether the EVs that are shed by *C. gattii* may be responsible for triggering the long-distance proliferation described above, we infected the non-outbreak strain ICB180 into macrophages and then added 10 μg of EVs, isolated from the outbreak strain R265, to the media. Exposing ICB180 to EVs 1 h prior to infection did not alter subsequent IPR. However, pre-treating macrophages with EVs before infection led to a small but significant increase in subsequent IPR and this effect was dramatically enhanced, in a dose-dependent manner (Fig. 3c), if EVs were added to the cells once infection was already established (Fig. 3a−c).

It has been reported previously that albumin presented in serum rapidly destabilises EVs[15]. We therefore considered whether the presence of serum proteins (in the pooled human serum (PHS) used to opsonise cryptococci prior to macrophage infection) might be responsible for the limited impact of EVs on cryptococci if they were added prior to infection. In agreement with that, repeating this experiment using the monoclonal antibody 18B7[16] (which specifically recognises capsular polysaccharide glucuronoxylomannan) instead of serum opsonisation led to a significantly increased IPR during subsequent infection (*P* = 0.0039, Wilcoxon paired *t* test; Fig. 3b, Ab opsonisation). Thus EVs act to trigger rapid proliferation in cryptococci but this activity can be inactivated by the presence of human serum.

Previous studies have shown that EVs isolated from *C. neoformans* and *Candida albicans* facilitate their phagocytosis into an intracellular niche[17,18]. However, the percentage of phagocytes containing engulfed ICB180-mCherry was similar in the presence or absence of EVs$_{R265}$ (*P* = 0.1114, Wilcoxon paired *t* test; Supplementary Fig. 4a) or when R265 yeasts were present in the transwell assay (*P* = 0.25, Wilcoxon paired *t* test; Supplementary Fig. 4a, ICB180$^{(+R265)}$). Moreover, the number

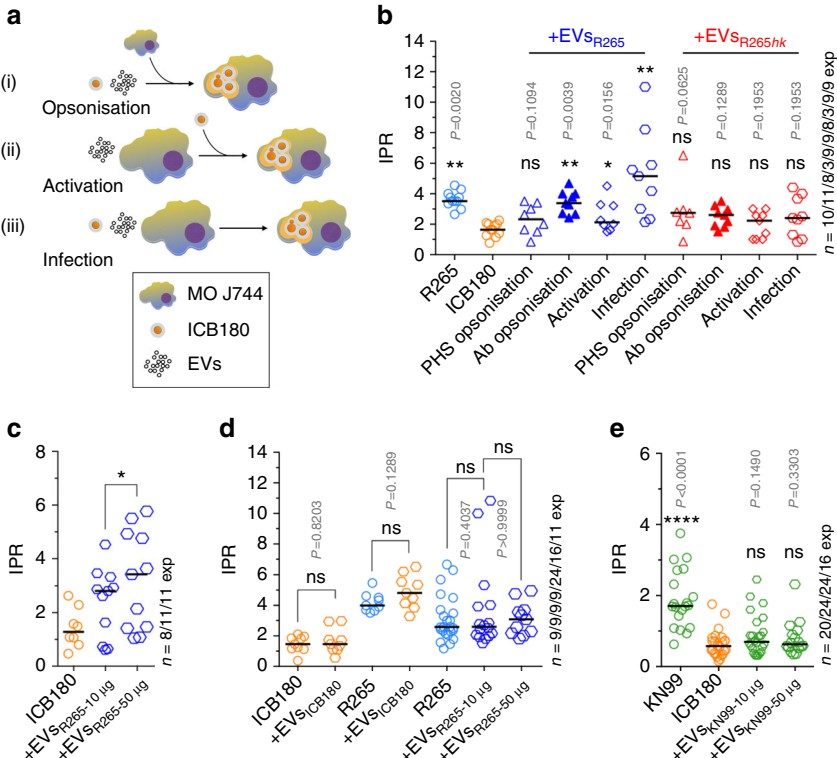

**Fig. 3** EVs increase survival of cryptococci inside macrophages. **a** A schematic representation of the experimental assay in which extracellular vesicles (EVs) were added at different time points, either (i) to the cryptococci during opsonisation, 1 h prior to infection ('Opsonisation'), (ii) directly to the macrophages J774 (MO J774) during 1 h of activation prior to infection ('Activation') or (iii) at the same time as the cryptococci are added to the macrophages ('Infection'). **b** IPRs of R265 growing alone (R265), ICB180 growing alone (ICB180) and in the presence of 10 µg of EVs isolated from R265 cells (EVs_{R265}) or heat-inactivated EVs_{R265} (EVs_{R265hk}) added at different stages of infection, as described above: during yeast opsonisation by pooled human serum ('PHS opsonisation') or by GXM-specific antibodies—Mab 18B7 ('Ab opsonisation'), J774 activation ('Activation') or during incubation with both macrophages and ICB180 yeast cells ('Infection'). Data are presented as scattered dot plots with lines representing their medians. Data are representative of results from 3 (with biological triplicates) to 11 independent experiments with 213–1767 total number of yeasts counted for each sample. Wilcoxon paired $t$ test where * ($P ≤ 0.05$), significant difference; ** ($P ≤ 0.01$; and $P = 0.0078$ for infection with EVs_{R265}), highly significant difference and ns ($P > 0.05$), not significantly different. **c** IPR values can be further increased after adding higher amounts of EVs (+EVs_{R265-50 µg}). Data are presented as scattered dot plots with lines representing their medians. Data are representative of results from 8 to 11 independent experiments with 1504–1948 total number of yeasts counted for each sample. Wilcoxon paired $t$ test where * ($P = 0.0186$), significant difference. **d** EVs isolated from ICB180 (+EVs_{ICB180}) do not increase IPRs of ICB180 or R265 and proliferation of R265 is not enhanced by its own EVs (+EVs_{R265}) even at higher concentration (+EVs_{R265-50 µg}). Data are presented as scattered dot plots with lines representing their medians. Data are representative of results from 9 to 24 independent experiments with 275–7888 total number of yeasts counted for each sample. Wilcoxon paired $t$ test where ns ($P > 0.05$), not significantly different. **e** IPR of a non-virulent ICB180 strain is not altered by the presence of EVs isolated from *C. neoformans* virulent strain KN99 even at higher concentration of those vesicles (+EVs_{KN99-50 µg}) added during the infection step. Data are presented as scattered dot plots with lines representing their medians. Individual Wilcoxon matched-pairs signed rank test presented as $P$ values above each dot plot, where ns ($P > 0.05$), not significantly different. Data are representative of results from at least 16 independent experiments with 1742–5209 total number of yeasts counted for each sample

of phagocytosed ICB180 yeasts per macrophage was not altered in the presence of native or heat-inactivated EVs_{R265} ($P = 0.8742$, Kruskal–Wallis test; Supplementary Fig. 4b). The majority of infected macrophages contained only one fungal cell, and we never observed more than three cells phagocytosed within first two hours of fungal infection (Supplementary Fig. 4b). Thus the increased IPR values we observe in the presence of R265-derived EVs do not result from a higher rate of phagocytosis.

Interestingly, only EVs from virulent, outbreak cryptococci appear to be able to induce this effect, since EVs isolated from ICB180 were not able to increase the IPR of ICB180 or R265 (Fig. 3d, +EVs_{ICB180}). Similarly, exposing R265 to additional EVs derived from the same strain could not further augment intracellular proliferation (Fig. 3d, +EVs_{R265-10 µg}) even at higher vesicle concentration (Fig. 3d, +EVs_{R265-50 µg}). EVs isolated from the outbreak strain were also not capable of boosting the intracellular survival of completely avirulent *C. gattii* strains CBS

8684 and NIH 312 (Supplementary Fig. 6). Lastly, EVs isolated from the virulent *C. neoformans* KN99 strain were unable to raise the IPR of ICB180 (Fig. 3e). Thus, taken together, these data suggest that EVs act as an 'accelerator' of intracellular proliferation (and thus virulence) within *C. gattii*, but only for strains that already have the capacity for low (but not zero) rates of intracellular proliferation.

**Enhanced survival requires proteins and RNA from outbreak EVs.** Heat inactivation of EVs_{R265} blocked their ability to trigger enhanced survival of ICB180 cells when added either before or during macrophage infection (Fig. 3b, +EVs_{R265hk}), suggesting that the activity of EVs requires either a heat-labile component or an intact vesicular structure that is rendered inactive by high temperature. To further test if capsule polysaccharides might have a role in promoting proliferation of yeasts inside macrophages, we

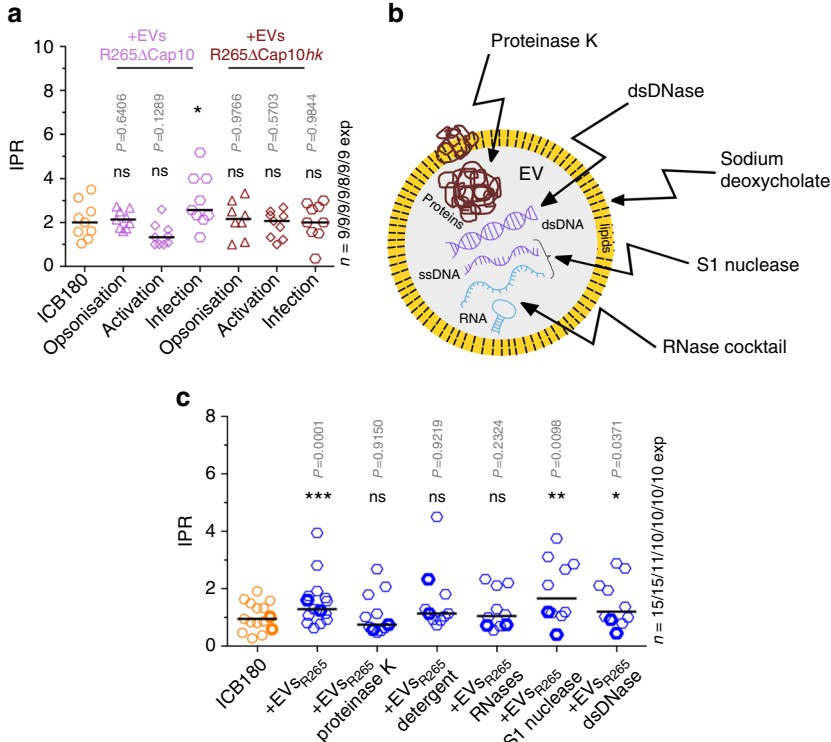

**Fig. 4** EV proteins and RNA are necessary to increase survival of cryptococci inside macrophages. **a** IPRs of ICB180 growing alone (ICB180) and in the presence of 10 μg of EVs isolated from acapsular strain R265ΔCap10 (EVs$_{R265ΔCap10}$) or heat-inactivated EVs$_{R265ΔCap10}$ (EVs$_{R265ΔCap10hk}$) added at different stages of infection: during yeast opsonisation using PHS (opsonisation), J774 activation (activation) or during incubation with both macrophages and ICB180 yeast cells (infection; see also Fig. 3a). Data are presented as scattered dot plots with lines representing their medians. Data are representative of results from 8 to 9 independent experiments with 147–301 total number of yeasts counted for each sample. Wilcoxon paired $t$ test where * ($P = 0.0117$), significant difference; and ns ($P > 0.05$), not significantly different. **b** Schematic drawing of the EV and treatments performed towards protein degradation via proteinase K, lipids degradation via sodium deoxycholate, double-stranded DNA (dsDNA) degradation via dsDNase, single-stranded DNA (ssDNA) and single-stranded regions of RNA degradation via S1 nuclease and further RNA degradation, including RNA duplexes, via RNase cocktail of RNase A and T1. **c** IPR values of ICB180 are increased in the presence of 10 μg of EVs (or 50 μg—symbols with thicker borders) isolated from R265 (+EVs$_{R265}$), EVs$_{R265}$ treated with S1 nuclease (+EVs$_{R265}$ S1 nuclease) and EVs$_{R265}$ treated with dsDNase (+EVs$_{R265}$ dsDNase) but not when EVs treated with proteinase K (+EVs$_{R265}$ proteinase K), sodium deoxycholate (+EVs$_{R265}$ detergent) or RNase cocktail (+EVs$_{R265}$ RNases) were used. Data are representative of results from 10 to 15 independent experiments with 1181–2691 total number of yeasts counted for each sample. Wilcoxon paired $t$ test where * ($P \leq 0.05$), significant difference; ** ($P \leq 0.01$), significant difference, *** ($P \leq 0.001$), significant difference and ns ($P > 0.05$), not significantly different

used EVs isolated from acapsular strain of R265, R265ΔCap10 (Fig. 4a, +EVs$_{R265ΔCap10}$) and their heat-inactivated form (Fig. 4a, +EVs$_{R265ΔCap10hk}$). Although heat-inactivated acapsular EVs were incapable of triggering increased IPR under any conditions, intact EVs$_{R265ΔCap10}$ showed a small but significant ability to increase ICB180 proliferation rates when they were present throughout the infection assay ($P = 0.0117$, Wilcoxon paired $t$ test; Fig. 4a, infection). This relatively weak effect presumably explains why R265ΔCap10 is unable to enhance the IPR of a non-outbreak strain in the transwell system (Fig. 2b), since the EV concentration in this transwell experimental setup is much lower than that used when supplementing with purified EVs. Thus, while the presence of capsule associated with the EVs significantly augments IPR, capsular polysaccharides may not be completely essential for the EV-mediated survival effect.

Next we set out to establish which EV components might be responsible for triggering augmented IPRs (Fig. 4b and Supplementary Fig. 5) by selectively degrading different classes of molecule previously shown to be enriched in EVs from other species. Removing double-stranded or single-stranded DNA from within the EVs did not impair their ability to enhance the IPR of ICB180 cells (Fig. 4c, +EVs$_{R265}$ $_{dsDNase}$, +EVs$_{R265}$ S1 nuclease; see also Supplementary Fig. 5c). However, removing proteins (Fig. 4c,

+EVs$_{R265}$ proteinase K; see also Supplementary Fig. 5d) or RNA (Fig. 4c, +EVs$_{R265}$ RNases; see also Supplementary Fig. 5c) or perturbing the lipid composition of the EV with non-denaturing sodium deoxycholate (Fig. 4c, +EVs$_{R265}$ detergent; see also Supplementary Fig. 5a and 5b) all eliminated the ability of EVs to trigger increased IPR in ICB180. Thus to functional effectively in raising IPR, EVs must be intact and contain protein and RNA, but not DNA, cargo.

**Fungal EVs are rapidly taken up by murine macrophages**. Our observations suggest that EVs derived from *C. gattii* are able to enter infected macrophages and impact on the rate of fungal intracellular proliferation within the phagosome. To test this hypothesis, we exposed macrophages to EVs from R265 and then fixed and stained for cryptococcal polysaccharide, which is a major EV cargo in cryptococci[19],[15] but absent from mammalian EVs. Cryptococcal EVs were rapidly internalised into macrophages (Fig. 5a, b), with a strong signal observed within 30 min of incubation and a more central localisation observed after 60−120 min (see also Supplementary Movie 1). Time-course analysis showed classical uptake kinetics, with a half time to peak internalisation of 17.5 min (Fig. 5c).

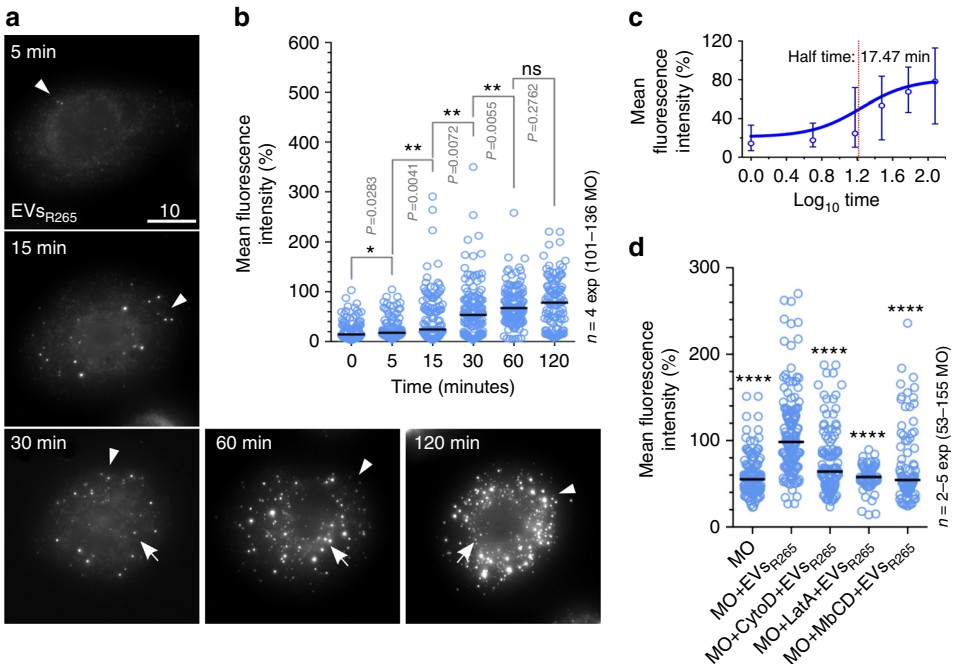

**Fig. 5** EVs are rapidly and actively internalised by macrophages. **a** Macrophages were exposed to 10 μg of EVs$_{R265}$, fixed with 4% PFA after 5, 15, 30, 60 and 120 min incubation and then immunostained using capsule-specific monoclonal antibody MAb 18B7. Images are maximum projections (21 z-stacks with 0.5 μm intervals). Note the change in signal from the cell periphery (arrowheads) to the cell body (arrows) in a time-dependent manner. Bar: 10 μm. **b** EV uptake increases in a time-dependent manner. Data are presented as scattered dot plots with lines representing their medians. Mean fluorescent intensities were normalised using the median value at 2 h as 100%. Data are representative of results from four independent experiments with a minimum of 25 macrophages analysed per sample per experiment. Unpaired Mann−Whitney tests where * ($P \leq 0.05$), significant difference; ** ($P \leq 0.01$), significant difference; and ns ($P > 0.05$), not significantly different. **c** Half time determination of EVs uptake by J774 macrophages. Data are presented as a sigmoidal non-linear regression curve (four-parameter logistic curve) with medians and errors (interquartile range) obtained from four independent experiments (from Fig. 5b). **d** The uptake of EVs by J774 macrophages is blocked by actin polymerisation inhibitors, cytochalasin D (MO+CytoD+EVs$_{R265}$) and latrunculin A (MO+LatA+EVs$_{R265}$) or a lipid raft-specific inhibitor methyl-β-cyclodextrin (MO+MbCD+EVs$_{R265}$), which depletes cholesterol. Data are presented as scattered dot plots with lines representing their medians. Graph showing percentages of mean fluorescent intensities from macrophages alone (MO) or incubated with MAb 18B7-immunostained EVs isolated from R265 (MO+EVs$_{R265}$) and were normalised to a median value obtained for MO +EVs$_{R265}$. Data are representative of results from 2 to 5 independent experiments with a minimum of 25 macrophages analysed per sample per experiment. Unpaired Mann−Whitney tests where **** ($P \leq 0.0001$), highly significant difference

A possible complication of this approach is that cryptococcal capsule is shed both within EVs and as free polysaccharide. To discriminate between these two sources, we performed imaging on cells permeabilised with the detergent Triton X-100 at either 0.1% (to permeabilise the macrophage plasma membrane) or 1% (to permeabilise all membranes, including those of EVs; Supplementary Fig. 7a). Using this approach we were able to demonstrate that overall capsular material detected was the same in both conditions when free capsular polysaccharide was added to the macrophages, but significantly higher in the 1% detergent conditions when EVs are added to the cells ($P < 0.0001$, Mann −Whitney test, Supplementary Fig. 7b). Thus the fluorescence observed in the images above is derived primarily from delivered EVs and not free polysaccharide.

We wondered whether fungal EVs were freely diffusing across the macrophage membrane or were actively taken up by host cells. Treatment of macrophages with the actin polymerisation inhibitors cytochalasin D and latrunculin A, or the cholesterol-depleting agent methyl-β-cyclodextrin, all significantly reduced EV uptake into macrophages (Fig. 5d), suggesting that EV uptake is an active, rather than passive, process. Surprisingly, the uptake of capsule was not affected in the presence of the methyl-β-cyclodextrin (Supplementary Fig. 8), implying a different route of uptake for free capsular elements in comparison to capsule contained within EVs.

**EVs colocalise with fungal cells inside host phagosomes**. Lastly, having established that extracellularly added EVs can be internalised by macrophages, we set out to test whether these vesicles can reach the phagosome and interact with engulfed cryptococci in order to modulate yeast survival. To avoid contamination with EVs released by the phagocytosed fungal cells themselves, we infected macrophages with the acapsular strain R265ΔCap10 expressing GFP (Fig. 6a, R265ΔCap10) and then added EVs isolated from wild-type (capsular) R265 (Fig. 6a, EVs$_{R265}$). Thus, only exogenously added EVs can be visualised by anti-capsular antibody 18B7. Within 15 min incubation, exogenously added EVs showed a strong colocalisation with acapsular cryptococci within the host phagosome (Fig. 6a, b). Interestingly, heat-inactivated EVs also localised rapidly to phagosomes (Fig. 6a, EVs$_{R265hk}$), although they are unable to induce raised IPR in intracellular cryptococci (Fig. 3b), suggesting that delivery to the phagosome is independent of functionality.

## Discussion

Teamwork between individuals presents an evolutionary conundrum, since most collaborative systems in biology are vulnerable to the evolution of 'cheaters' and are thus unstable. This is particularly true of infectious organisms, in which selection by the host is intense, and yet 'division of labour'

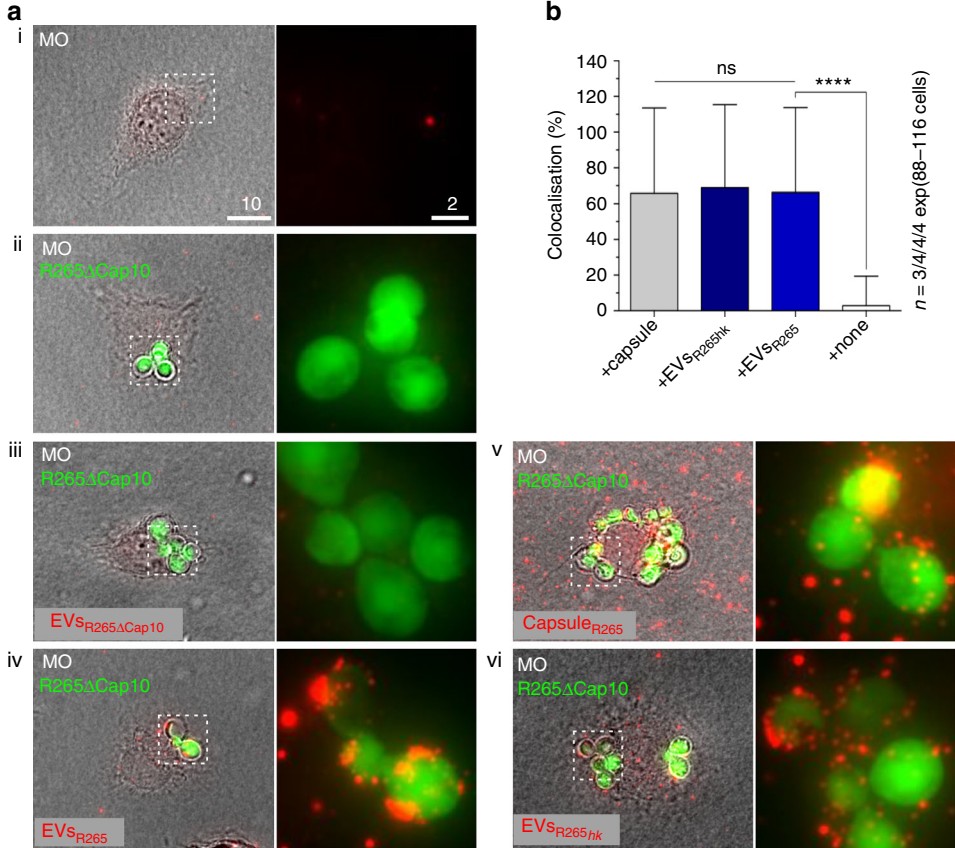

**Fig. 6** EVs added to macrophages infected with acapsular R265-GFP rapidly colocalise with the cryptococcal phagosome. **a** Macrophages alone (MO; i), macrophages with phagocytosed R265ΔCap10-GFP cells (ii) or macrophages containing R265ΔCap10-GFP and co-incubated with EVs isolated from R265ΔCap10 (EVs$_{R265ΔCap10}$; (iii)) are not recognised by capsule-specific MAb 18B7. Those macrophages co-incubated with EVs isolated from R265 (EVs$_{R265}$; (iv)), with capsule isolated from R265 (Capsule$_{R265}$; (v)) or heat-inactivated EVs isolated from R265 (EVs$_{R265hk}$; (vi)) show colocalisation between R265ΔCap10-GFP cells and red signal suggesting that the EVs and capsule polysaccharides were delivered to the phagosomes. Pictures represent maximum intensity projections of 21 z-stacks obtained from 10 μm cross section through macrophages. Bars: 10 and 2 μm. **b** Graph showing percentage of colocalised signals of GFP labelled yeast cells R265ΔCap10 engulfed by murine macrophages and MAb 18B7-immunostained capsule (+capsule), heat-inactivated EVs isolated from R265 (EVs$_{R265hk}$) or EVs isolated from R265 (+EVs$_{R265}$). As a negative control intensities from macrophages infected with R265ΔCap10-GFP without EVs were used (+none). All numbers are given as means ± standard deviation (s.d.) and are representative of results from 3 to 4 independent experiments with a minimum of 25 macrophages analysed per sample per experiment. ns indicate statistically non-significant difference (Kruskal−Wallis test, $P = 0.89$) and quadruple asterisks indicate a highly significant difference (****, unpaired Mann−Whitney test, $P ≤ 0.0001$)

mechanisms of virulence appear to have evolved independently in unrelated organisms such as *Salmonella typhimurium* and *C. gattii*[20]. Understanding how such systems are coordinated at the cellular level is a key challenge for understanding how they may have evolved. In this report, we provide evidence that 'division of labour' in cryptococci occurs remotely, through release of EVs by the Pacific Northwest outbreak strain. These vesicles diffuse over significant distances, become rapidly internalised by macrophages and then trigger increased rates of proliferation in cryptococci residing within the phagosome.

Fungal EVs have previously been shown to transmit a proinflammatory message to macrophages[17,18,21], but to the best of our knowledge, have not previously been implicated in fungal virulence. Our data indicate that virulence enhancement by EVs requires the presence of cryptococcal capsule associated with the EVs, but that capsular material alone is insufficient to recapitulate the phenotype. Detailed analysis of EV contents from a range of pathogenic fungi, including cryptococci, has revealed a wide spectrum of RNAs, short non-coding mRNAs, ribosomal proteins and proteins linked to virulence, anti-oxidant defence and pathogenesis that may facilitate yeast survival and proliferation[18,22−29]. Our data suggest that RNA and proteins protected by

lipid bilayer are necessary to transfer the virulence in EVs. Future studies to characterise C. gattii outbreak EVs in more detail and determine how their contents trigger raised intracellular proliferation in recipient cells are therefore likely to be of considerable value in understanding how microbes communicate over substantial distances in order to coordinate virulence approaches.

## Methods

**Yeast strains, cell lines, culture media and growth conditions.** All reagents were purchased from Sigma unless otherwise stated. *C. gattii* and *C. neoformans* strains (Supplementary Table 1) and murine macrophage-like cell line J774A.1 (mouse BALB/cN; ATCC® TIB-67™) were cultured as described previously[7]. In order to obtain fluorescently labelled strains biolistic transformation was performed[7,30] using a plasmid pAG32_GFP[7] for R265ΔCap10 and a plasmid pRS426H-CnmCherry[31] for ICB180.

**Validation of stable transformants.** For selection of stable transformants three independent experiments were carried out as described previously[31]. Growth of ICB180 and ICB180-mCherry strains was performed in 24-well plates (Greiner) in liquid yeast-peptone-dextrose (YPD; 2% glucose, 1% peptone and 1% yeast extract) medium shaking at 200 revolutions per minute (rpm) at 37 °C at the starting concentration of $0.5×10^6$ cells per ml and was monitored by optical density at a wavelength of 600 nm (OD600) measurements by a plate reader FLUOstar Omega.

Linear regressions were calculated using GraphPad Prism Software Inc. Cell size of ICB180 and ICB180-mCherry refers to the yeast diameter without its capsule and was measured from 24-h YPD cultures using Fiji software[32] after acquiring pictures on Nikon TE2000-U live microscope equipped with Digital Sight camera (DS-Qi 1MC) and NIS elements AR software and ×100 oil immersion lens.

**Isolation of cryptococcal capsule**. Capsular purification was adapted from Cherniak et al.[33] and slightly modified as follows: R265 cells grew in 500 ml YPD liquid culture at room temperature with continuous slow shaking (40 rpm). After 10 days growth, the culture was autoclaved (127 °C, 35 min) and cells were removed via centrifugation (10,000 × $g$ for 10 min), followed by filtering of the supernatant through 0.45 μm membranes (GE Healthcare Life Sciences #7184-004) to ensure only cell-free supernatant remained. Precipitation of the capsular poly-saccharides from the supernatant was performed by adding three volumes of cold ethanol followed by centrifugation to pellet the capsule (12,000 × $g$ for 15 min). The pellet was washed twice in cold ethanol and vacuum dried over night. Dried capsule was kept at 4 °C and dissolved in PBS to the stock concentration of 1 mg ml$^{-1}$ which was kept at −20 °C.

**Isolation and characterisation of EVs**. Cryptococcal EVs were isolated by dif-ferential centrifugation according to Rodrigues et al.[13]. Briefly, *C. gattii* cells were grown in 30 ml YPD cultures for 48 h slowly shaking (50–100 rpm) at 25 °C, then 15 ml was transferred to 500 ml YPD cultures for 72 h vigorously shaking (180 rpm) at 25 °C. Supernatant was separated from cells in initial centrifugation at 4000 × $g$ for 10–15 min and transferred to 250 ml vessels for the following two-step centrifugation at 5000 × $g$ for 15 min and subsequent 12,000 × $g$ for 20 min at 4 °C using rotor JLA 16.250 and Avanti JXN-26 high-speed centrifuge system (Beckman Coulter). Obtained supernatants were transferred through Whatman 0.8 μm membranes (GE Healthcare Life Sciences #7188-004) using a vacuum pump and later through 0.45 μm membranes (GE Healthcare Life Sciences #7184-004) or gently through Acrodisc® Syringe Filters with Supor® Membrane (Pall Life #4654). The resulting flow through was concentrated using Amicon-Ultra columns cut off 100 kDa (Millipore #UCF910008). Concentrated solutions were ultracentrifuged at 100,000 × $g$ for 1 h at 4 °C using Optima ultracentrifuge XPN-80 and rotor 70 Ti (Beckman Coulter). Pelleted vesicles were resuspended in 200 μl of filtered PBS or filtered sterile water (for TEM) and checked by plating onto YPD agar for the presence of live cells. EV preparations were tested for protein concentration using Micro BCA Protein Assay Kit (Thermo Fisher Scientific #23235) and were stored at −20 °C until further use. Particle diameter for each isolation was measured in duplicates using the Particle Size Analyser Insight NanoSight (1450/118) and prism NTA4000 LM10 Optical Flat (Malvern) with camera shutter 1495 and gain of 400. Size distribution histograms were created using GraphPad Prism Software Inc. Transmission electron microscope was performed as described previously[34]. Where necessary, EVs were heat inactivated at 60 °C for 1 h.

**Infection assays, transwells, IPR, phagocytic index**. Macrophages (0.5–1.0×10$^5$ cells per ml) were seeded into a 24-well plate (Greiner) in a low-glucose (1000 mg l$^{-1}$) Dulbecco's modified Eagle's medium (DMEM) supplemented with 2 mM L-glutamine, 100 U ml$^{-1}$ penicillin, 100 U ml$^{-1}$ streptomycin and 10% foetal bovine serum (FBS) at 37 °C and 5% CO$_2$ and after 18–24 h they were activated for 1 h with 150 ng ml$^{-1}$ phorbol 12-myristate 13-acetate (PMA; #P8139) in serum-free DMEM (SF-DMEM). During that time 10$^7$ yeasts per ml were opsonised with 5% PHS or 10 μg ml$^{-1}$ MAb 18B7 for 1 h at room temperature. Phagocytes were infected with opsonised cryptococci (0.5–1.0×10$^6$ cells per ml; MOI 10:1) for 2 h at 37 °C and 5% CO$_2$. For co-infections with transwell filter system (Thincert, Greiner Bio-One #662641) top compartment contained 0.5–1.0×10$^6$ opsonised yeasts resuspended in 350 μl of SF-DMEM to the total volume of 400 μl. Transwells were removed after 2 h of infections. After 2 h ($T_0$) and following 24 h of infection ($T_{24}$) IPR measurements were performed as described previously[6]. Recovery of the capsule by the acapsular strain R265ΔCap10 was performed at room temperature for 1 h, where 0.5×10$^6$ cells were mixed with 12.5 μg of the capsule isolated from R265.

Phagocytic index was scored microscopically within 2 h of infection as a number of macrophages containing one or more cryptococci in the absence or in the presence of 10 μg of EVs. Number of yeast cells phagocytosed by macrophages was calculated as the number of individual ICB180 cells inside the infected phagocytes within 2 h of infection, where budded but non-detached cells were counted as single cells. For co-infection studies, R265_GFP6[7] and ICB180-mCherry strains were used at a ratio of 1:1 and the concentration of 0.5×10$^6$ cells per ml and the growth was performed in 48-well plates (Greiner) seeded with 0.5×10$^5$ cells per ml macrophages as described above. After 2- or 24-h of infection brightfield and fluorescent images were taken using Nikon TE2000-U live microscope with Digital Sight camera (DS-Qi 1MC) and NIS elements AR software.

**EV treatments**. To remove proteins from the EVs, 50 μl of EVs$_{R265}$ (at protein concentration of around 6 mg ml$^{-1}$) was gently mixed with proteinase K from *Tritirachium album* (at final 100 μg ml$^{-1}$ concentration; #P4850) and incubated for 1.5 h at 37 °C. Proteinase K was then inactivated by addition of 1 mM phenylmethanesulfonyl fluoride (#93482) and transferring the tube to room

temperature for 30 min. The reduction of protein levels after proteinase K treat-ment was confirmed by running 10 μg of untreated and proteinase K-treated EVs$_{R265}$ on a 4–20% Mini-PROTEAN® TGX Stain-Free™ Protein Gel (Bio-Rad #4568095) using PageRuler™ Plus Prestained Protein Ladder, 10−250 kDa (Thermo Fisher Scientific #26619).

To remove lipids, 50 μl of EVs$_{R265}$ (at protein concentration of around 6 mg ml$^{-1}$) was incubated in the presence of 0.25% sodium deoxycholate (resuspended in PBS) for 24 h at 4 °C. The reduction of lipids was confirmed by staining detergent-treated EVs$_{R265}$ with Vybrant DiI[35] (see below).

To remove double-stranded DNA, 0.4 μl dsDNase (Thermo Fisher Scientific #EN0771) was added to 20 μl (at protein concentration of around 6 mg ml$^{-1}$) EVs$_{R265}$ for 15 min at 37 °C.

To degrade single-stranded DNA and RNA deprived of double-stranded regions, 0.4 μl of S1 nuclease (Thermo Fisher Scientific #EN0321) was added to 20 μl EVs$_{R265}$ for 30 min at room temperature.

To degrade RNA, 50 μl of EVs$_{R265}$ (at protein concentration of around 6 mg ml$^{-1}$) was incubated in the presence of 2.5 μl RNase cocktail of RNase A and T1 (RNase Cocktail™ Enzyme Mix, Thermo Fisher Scientific #AM2286) for 15 min at 37 °C. Degradation of nucleic acids was confirmed by isolation of nucleic acids from treated and untreated EVs$_{R265}$ using Wizard Genomic DNA Purification Kit (Promega) without RNase A added and visualising them on 1% agarose gel.

All treated EVs were used immediately in IPR studies or were stored at −20 °C until further use.

**Vybrant DiI staining and immunostaining**. Cryptococci or infected macrophages that were grown on 13 mm-diameter coverslips (VWr #631-0150) in 24-well plates underwent fixation using pre-warmed 4% paraformaldehyde for 10 min at room temperature, washed in triplicate in warmed PBS and then treated with 50 mM NH$_4$Cl (Fluka) to quench fixation followed by three washes with PBS. After that cells were treated with 0.1 or 1% Triton X-100 (in PBS) for 5 min at room temperature. Wells were washed in PBS in triplicate and cells were blocked using goat serum (#G9023) for 30–60 min at room temperature. Wells were washed in PBS in triplicate and incubated with primary antibodies MAb 18B7 at the concentration 1:500 (20 μg ml$^{-1}$) for 30–60 min at room temperature. Wells were washed in PBS in triplicate and incubated in dark with goat anti-mouse IgG conjugated with CF594 at the concentration 1:500 (#SAB4600402) for 30–60 min at room temperature. Wells were washed in PBS in triplicate and the coverslips were transferred onto poly-L-lysine adhesive microscope slides (Grale HDS, Trajan Scientific) and mounted using ProLong Gold Antifade Mountant (Thermo Fisher Scientific #P36934).

EVs$_{R265}$ and detergent-treated EVs$_{R265}$ were stained with 20 μM (final concentration) of a fluorescent lipophilic dye 1,1′-Dioctadecyl-3,3,3′,3′-Tetramethylindocarbocyanine Perchlorate (Vybrant DiI cell-labeling solution, Molecular Probes) for 105 min at room temperature followed by 15 min at 37 °C. Unbound dye was removed from the samples by washing twice in 14 ml PBS and filtering the resuspended samples through Amicon columns with a cut off of 100 kDa. Labelled EVs were then incubated for 30 min with macrophages, washed three times with warmed PBS and fixed with 4% paraformaldehyde as described above and observed microscopically.

**Inhibition of endocytosis**. To inhibit endocytosis, cytochalasin (#C2618-200ul) was used at the final concentration of 5 μM (in DMSO) and latrunculin A (Cal-biochem #428026-50UG) at 2 μM. These inhibitors were added to macrophages 1 h prior to addition of EVs. Methyl-β-cyclodextrin (#C4555-1G) at the final con-centration of 5 mM (in PBS) was added to macrophages 30 min prior to adding the EVs. After initial inhibition, macrophages were washed with warmed PBS and incubated with 10 μg of EVs or 1–10 μg of capsule for 15 min in SF-DMEM.

**Microscopy**. Infected/immunostained macrophages were observed using Zeiss Axio Observer equipped with a ORCA-Flash 4.0 camera and ×63 oil immersion lens. Twenty-one z-stacks with 0.5 μm intervals were acquired at 150 ms exposure time for red fluorescent signals (100% lamp intensity) and 100 ms exposure time for GFP signals (20% lamp intensity). Maximum projections were performed using Zeiss software or Fiji[32]. The mean or maximum pixel intensities of red fluorescent signals derived from immunostained EVs were corrected for the adjacent back-ground outside the macrophage, and additionally in co-localisation studies between EVs and phagocytosed yeasts, for the adjacent background in the macrophage.

**Statistical analysis**. All statistical analyses were performed using GraphPad Prism software (GraphPad Software Inc.).

**Data availability**. The data that support the findings of this study are available in this article and its Supplementary Information files, or from the corresponding authors upon request.

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

## Acknowledgements

We are grateful to Peter Williamson (Laboratory of Clinical Infectious Diseases, NIAID, NIH, Bethesda, Maryland, USA) for providing the pKUTAP plasmid and to Arturo Casadevall (Department of Microbiology and Immunology, Albert Einstein College of Medicine of Yeshiva University, New York, USA) for mouse IgG 18B7. We would like to thank HAPI lab members for fruitful discussions and Mrs Jude Williams for overall administration of the MITOFUN project. The research leading to these results has received funding from the European Research Council under the European Union's Seventh Framework Programme (FP/2007-2013)/ERC Grant Agreement No. 614562 and R.C.M is additionally supported by a Wolfson Royal Society Research Merit Award.

## Author contributions

E.B. conceived, designed and directed the project, performed most experiments, analysed the data and assembled all figures. M.A.S. performed EV isolations, IPR measurements with EVs and analysed the data. M.A. performed co-infections between R265-GFP and ICB180-mCh and analysed the data. C.B. tested survival of ICB180-mCh under stress conditions. E.J.O'D. conducted inhibition of endocytic uptake. E.B. and R.C.M. wrote the manuscript. Feedback was provided from M.A.S., M.A. and E.J.O'D.

## Additional information

**Competing interests:** The authors declare no competing interests.

