## [Peer Review File · Nature Communications]

Reviewers' comments:

Reviewer #1 (Remarks to the Author):

Bielska and colleagues investigated the role of *Cryptococcus gattii* extracellular vesicles in the control of virulence phenotypes. The manuscript is straightforward, extremely well written and brings novel information to the literature at multiple levels. First, through the use of a fungal strain associated with an important outbreak of cryptococcosis, the manuscript suggests an intriguing mechanism by which virulence mechanisms could be propagated within fungal populations. Most importantly, the manuscript brings to the literature the first concrete demonstration that fungal EVs impact virulence. The recent literature repetitively suggests that fungal EVs are important for virulence – they are actually called ‘virulence bags’ by many authors – but they have never been directly associated with virulence phenotypes. The authors designed an interesting model to prove the concept that fungal EVs are in fact virulence-related compartments. I have a few comments that may be useful for improving the manuscript:

1. Please avoid using ‘secretion’ and related terms (lines 51, 78, 95, 207 and legend for Figure 1) when referring to fungal EVs. It is still unknown if the formation of fungal EVs results from secretory events – there is evidence in the literature suggesting that they can originate from excretion or just regulation of cell volume through cytoplasmic subtraction, among other possibilities. It is always safer to use “release and exchange of extracellular vesicles” than “secretion and exchange of extracellular vesicles”. The authors actually used the correct form in the Abstract, line 31 (Purified vesicles released...).
2. Similarly, avoid using “purified vesicles” (lines 31, 114, 171, legend for Supplementary Figure 2). The methods currently available for obtaining fungal EVs certainly result in the co-isolation of vesicles of very different cellular origins, since they are based on physical chemical properties of EVs rather than on their composition. Therefore, it is very unlikely that the EV preparations used in this and previous studies are actually purified.
3. I think using “large cellular distance” is very subjective (lines 30, 59, 79) since the actual distances were not evaluated in this manuscript.
4. Line 126, please make clear that mAb 18B7 recognizes cryptococcal GXM.
5. Heat inactivation versus biological activity of EVs. The authors demonstrate that biological activity of EVs require thermosensitive compounds, which in fact argues against the hypothesis that polysaccharides and pigments (usual EV components) are responsible for controlling virulence phenotypes. I think the authors need to go deeper in this specific assay to give the reader a more concrete idea of the vesicle compounds associated to their interesting findings. My suggestion is that, in addition to using intact, heat-inactivated and albumin-treated EVs, the authors perform detergent extraction to assess the role of lipids, protease treatment to evaluate the involvement of polypeptides, and nuclease treatment to check if RNA and DNA are required for the effects of EVs.
6. Is it possible that the phenomenon herein described is functional when different species are used? Can *C. neoformans* EVs modify virulence phenotypes of *C. gattii*?

Reviewer #2 (Remarks to the Author):

The manuscript by Bielsk et al. describes the function of extracellular vesicles (EVs) from virulent fungi *Cryptococcus gattii* in pathogen-to-pathogen communication during host infection. It appears that heat-labile component in EVs or physical structure of vesicles, not capsular materials, is a major factor involved in modulation of intracellular proliferation of *Cryptococcus gattii* in macrophages. I have a few comments for consideration.

1. Unlike wildtype R265, acapsular R265 strain was unable to induce increased proliferation of intracellular ICB180 in the transwell system (Fig. 2). However, addition of EVs isolated from acapsular strain R265 Δ Cap10 increased the IPRs of ICB180 (Fig. 4). The authors need to address the causes of differences.
2. The method used to isolate capsule involved autoclave before precipitation of capsular polysaccharides which might have unwanted impact on the isolated polysaccharide. Did the author try to isolate capsule without autoclave and determine its effect on proliferation rate of ICB180.
3. The authors showed that IPR value of ICB180 increased in the presence of R265-derived EVs but there was no significant difference in the rate of phagocytosis based on the phagocytic index (PI) value in the presence or absence of the EVs. The PI value is calculated by the number of macrophages which contains engulfed fungal cells regardless of the number of fungal cells in macrophages. The number of *Cryptococcus* cells engulfed in macrophages just after infection is important because small difference of the number of fungal cells at the initial time point can cause huge difference after further incubation for 24 hours. Therefore, additional figures showing the number of engulfed fungal cells at the initial time in the presence or absence of the EVs would be helpful.
4. The authors determined the effect of adding R265 Δ Cap10 and capsular material separately in figure 2. However, it is also important to determine the effect of combining R265 Δ Cap10 and capsular material in the same assay.
5. It would be helpful to have the data of "infection" and "opsonisation" with Mab18B7 and serum side by side in Fig. 4d for readers to understand the effect of serum.
6. The methods and results of EVs isolation were similar to that of the close species *C. neoformans*. Therefore, Fig. 3 should be moved to supplemental material. It would be interesting to know if the effect of R265 EVs is species specific. Did the authors try to use EVs of H99 for IPR value of ICB180? Also, is the effect of R265 EVs strain specific or did the authors use different strains with less virulent other than ICB180?

Minor comments

1. In this study, two opsonization methods (by serum and Mab18B7) were used. Please indicate opsonization method in graph or figure legend to help readers.
2. There are several citations only have a number (line 248 and 261). Please spell out the first author for the source of references.
3. *Cryptococcus gattii* and *Cryptococcus neoformans* have to be italicized in Literature section.

Reviewers' comments:

Reviewer #1 (Remarks to the Author):

1. Please avoid using 'secretion' and related terms (lines 51, 78, 95, 207 and legend for Figure 1) when referring to fungal EVs. It is still unknown if the formation of fungal EVs results from secretory events – there is evidence in the literature suggesting that they can originate from excretion or just regulation of cell volume through cytoplasmic subtraction, among other possibilities. It is always safer to use “release and exchange of extracellular vesicles” than “secretion and exchange of extracellular vesicles”. The authors actually used the correct form in the Abstract, line 31 (Purified vesicles released...).

This is a very fair comment – we have now changed 'secretion/secreted' to 'release/released'.

2. Similarly, avoid using “purified vesicles” (lines 31, 114, 171, legend for Supplementary Figure 2). The methods currently available for obtaining fungal EVs certainly result in the co-isolation of vesicles of very different cellular origins, since they are based on physical chemical properties of EVs rather than on their composition. Therefore, it is very unlikely that the EV preparations used in this and previous studies are actually purified.

We have now changed to 'isolated vesicles' throughout the manuscript.

3. I think using “large cellular distance” is very subjective (lines 30, 59, 79) since the actual distances were not evaluated in this manuscript.

Revised to 'cellular distance'.

4. Line 126, please make clear that mAb 18B7 recognizes cryptococcal GXM.

This has been changed.

5. Heat inactivation versus biological activity of EVs. The authors demonstrate that biological activity of EVs require thermosensitive compounds, which in fact argues against the hypothesis that polysaccharides and pigments (usual EV components) are responsible for controlling virulence phenotypes. I think

the authors need to go deeper in this specific assay to give the reader a more concrete idea of the vesicle compounds associated to their interesting findings. My suggestion is that, in addition to using intact, heat-inactivated and albumin-treated EVs, the authors perform detergent extraction to assess the role of lipids, protease treatment to evaluate the involvement of polypeptides, and nuclease treatment to check if RNA and DNA are required for the effects of EVs.

This is a very fair comment. Consequently, we have now performed substantial additional experimentation, the data from which demonstrate that protein and RNA, but not DNA, components of the EVs are responsible for this effect. We have added these data as Figure 4 and Supplementary Figure 5.

6. Is it possible that the phenomenon herein described is functional when different species are used? Can *C. neoformans* EVs modify virulence phenotypes of *C. gattii*?

An interesting question. To answer it, we have now undertaken additional experiments using the *C. neoformans* KN99 strain. EVs isolated from KN99 strain were not able to increase proliferation of non-outbreak *C. gattii* strain, even at higher EVs concentration, suggesting that the effect is species-specific, and we now present these data in Figure 3e.

Reviewer #2 (Remarks to the Author):

The manuscript by Bielska et al. describes the function of extracellular vesicles (EVs) from virulent fungi *Cryptococcus gattii* in pathogen-to-pathogen communication during host infection. It appears that heat-labile component in EVs or physical structure of vesicles, not capsular materials, is a major factor involved in modulation of intracellular proliferation of *Cryptococcus gattii* in macrophages. I have a few comments for consideration.

1. Unlike wildtype R265, acapsular R265 strain was unable to induce increased proliferation of intracellular ICB180 in the transwell system (Fig. 2). However, addition of EVs isolated from acapsular strain R265 Δ Cap10 increased the IPRs of ICB180 (Fig. 4). The authors need to address the causes of differences.

We thank the reviewer for pointing out this potential confusion. The

explanation is one of dose – in the transwell assay, released EVs are at low concentration, whereas isolated EVs are much more concentrated. Indeed the effect we see with isolated EVs from R265ΔCap10 is very weak and consequently likely to be undetectable in the transwell assay. We have now added explanation of this to the text (lines 170-173).

2. The method used to isolate capsule involved autoclave before precipitation of capsular polysaccharides which might have unwanted impact on the isolated polysaccharide. Did the author try to isolate capsule without autoclave and determine its effect on proliferation rate of ICB180.

As suggested, we have now repeated these experiments without autoclaving and the results are the same. We now include these additional data as Supplementary Figure 2.

3. The authors showed that IPR value of ICB180 increased in the presence of R265-derived EVs but there was no significant difference in the rate of phagocytosis based on the phagocytic index (PI) value in the presence or absence of the EVs. The PI value is calculated by the number of macrophages which contains engulfed fungal cells regardless of the number of fungal cells in macrophages. The number of Cryptococcus cells engulfed in macrophages just after infection is important because small difference of the number of fungal cells at the initial time point can cause huge difference after further incubation for 24 hours. Therefore, additional figures showing the number of engulfed fungal cells at the initial time in the presence or absence of the EVs would be helpful.

This is a very fair comment. We have now counted the number of phagocytosed yeasts per macrophage and present these data, which show that variation in phagocytosis does not account for this observed effect, in Supplementary Figure 4.

4. The authors determined the effect of adding R265ΔCap10 and capsular material separately in figure 2. However, it is also important to determine the effect of combining R265ΔCap10 and capsular material in the same assay.

This is an interesting point. To address it we have combined both R265ΔCap10 and capsular material in the same assay (now shown in Figure 2b as ICB180^(+R265ΔCap10+R265 capsule)) and repeated this experiment using non-autoclaved

capsule (Supplementary Figure 2 as ICB180^(+R265ΔCap10 mixed with capsuleR265)). In both cases, combining both capsule and the acapsular strain at the same time are not sufficient to rescue the effect on IPR.

5. It would be helpful to have the data of “infection” and “opsonisation” with Mab18B7 and serum side by side in Fig. 4d for readers to understand the effect of serum.

We have modified the graph as suggested, which is now presented in Figure 3b.

6. The methods and results of EVs isolation were similar to that of the close species *C. neoformans*. Therefore, Fig. 3 should be moved to supplemental material. It would be interest to know if the effect of R265 EVs is species specific. Did the authors try to use EVs of H99 for IPR value of ICB180? Also, is the effect of R265 EVs strain specific or did the authors use different strains with less virulent other than ICB180?

We are grateful for these helpful suggestions. As suggested, previous Figure 3 has now been moved to Supplementary Figures as a Supplementary Figure 3. Regarding the second question – we have now tested *C. neoformans* KN99 strain (in response to Reviewer One’s suggestion) and EVs isolated from KN99 strain were not able to increase proliferation of non-outbreak *C. gattii* strain, even at higher EVs concentration. These data are presented in Figure 3e. Regarding the third question – we used two avirulent strains of *C. gattii*, CBS 8684 and NIH 312, but EVs isolated from R265 were not able to increase their proliferation in macrophages, even at higher vesicle concentration. These data are presented in Supplementary Figure 6 and discussed in the main text.

Minor comments

1. In this study, two opsonization methods (by serum and Mab18B7) were used. Please indicate opsonization method in graph or figure legend to help readers.

The graph was modified according to the above suggestion and is presented in Figure 3b.

2. There are several citations only have a number (line 248 and 261). Please

spell out the first author for the source of references.

The names were added.

3. *Cryptococcus gattii* and *Cryptococcus neoformans* have to be italicized in Literature section.

The names are italicized now.

REVIEWERS' COMMENTS:

Reviewer #1 (Remarks to the Author):

The manuscript is in great shape and essentially ready for publication. However, I need to point out that something really minor deserves attention. There is no evidence that fungal EVs are in fact exosomes, since it remains to be proved that they, in fact, derive from multivesicular bodies. Therefore, my final request is: please avoid using using "exosomes", " exosomal" and similar terms (lines 22, 177, legend for Figure 4, supplementary Figure 5). Please use just extracellular vesicles and it will be always correct.

Reviewer #2 (Remarks to the Author):

The authors have made many improvements in the revised manuscript. However, I have one major concern. The observations that combining both purified capsule and the acapsular R265 Δ Cap10 strain were not able to rescue the effect on IPR (Fig2), and yet the polysaccharide capsule was present in R265 Δ Cap10 in the experiment shown in Fig. S2 suggest that capsular deficiency was not the main cause for the inability of acapsular R265 to induce increased proliferation of ICB180 and instead EV of R265 Δ Cap10 had much reduced ability to increase ICB180 proliferation rate (line 169). Unless the authors have other evidence to further support their conclusion that capsular is necessary to induce higher intracellular proliferation rates, the subtitle and conclusion in lines 80 to 93, line 174 and line 249 regarding the role of capsule need to be modified.

Reviewers' comments:

Reviewer #1 (Remarks to the Author):

The manuscript is in great shape and essentially ready for publication. However, I need to point out that something really minor deserves attention. There is no evidence that fungal EVs are in fact exosomes, since it remains to be proved that they, in fact, derive from multivesicular bodies. Therefore, my final request is: please avoid using using “exosomes”, “exosomal” and similar terms (lines 22, 177, legend for Figure 4, supplementary Figure 5). Please use just extracellular vesicles and it will be always correct.

We agree with this comment and have changed all mentioned above to extracellular vesicles or EVs.

Reviewer #2 (Remarks to the Author):

The authors have made many improvements in the revised manuscript. However, I have one major concern. The observations that combining both purified capsule and the acapsular R265 Δ Cap10 strain were not able to rescue the effect on IPR (Fig2), and yet the polysaccharide capsule was present in R265 Δ Cap10 in the experiment shown in Fig. S2 suggest that capsular deficiency was not the main cause for the inability of acapsular R265 to induce increased proliferation of ICB180 and instead EV of R265 Δ Cap10 had much reduced ability to increase ICB180 proliferation rate (line 169). Unless the authors have other evidence to further support their conclusion that capsular is necessary to induce higher intracellular proliferation rates, the subtitle and conclusion in lines 80 to 93, line 174 and line 249 regarding the role of capsule need to be modified.

We agree that presented results might be confusing, as capsule can be shed extracellularly inside the EVs and as a free form. Experiments presented in Figure 2 and Figure S2 combined purified capsule and the acapsular R265 Δ Cap10 strain for a short period of time. This restores a small external

capsule to R265 Δ Cap10, but does not restore the capsular synthesis machinery and may not supply capsule to the inside of EVs. Overall, these results might suggest that presence of capsular material inside the EVs_{R265} increased the overall survival of ICB180 cells within macrophages in comparison to EVs_{R265 Δ Cap10}. However, we agree with the reviewer that the evidence we supply is indirect and therefore have modified the manuscript throughout to clarify this.